# Challenges and Opportunities in Cytopathology Artificial Intelligence

**DOI:** 10.3390/bioengineering12020176

**Published:** 2025-02-13

**Authors:** Meredith A. VandeHaar, Hussien Al-Asi, Fatih Doganay, Ibrahim Yilmaz, Heba Alazab, Yao Xiao, Jagadheshwar Balan, Bryan J. Dangott, Aziza Nassar, Jordan P. Reynolds, Zeynettin Akkus

**Affiliations:** 1Cytology, Department of Laboratory Medicine and Pathology, Mayo Clinic, Rochester, MN 55905, USA; vandehaar.meredith@mayo.edu; 2Computational Pathology and Artificial Intelligence, Department of Laboratory Medicine, Mayo Clinic, Jacksonville, FL 32224, USA; alasi.hussien@mayo.edu (H.A.-A.); doganay.fatih@mayo.edu (F.D.); yilmaz.ibrahim@mayo.edu (I.Y.); alazab.heba@mayo.edu (H.A.); dangott.bryan@mayo.edu (B.J.D.); nassar.aziza@mayo.edu (A.N.); reynolds.jordan@mayo.edu (J.P.R.); 3Computational Biology, Quantitative Health Science, Mayo Clinic, Rochester, MN 55905, USA; xiao.yao2@mayo.edu (Y.X.); balan.jagadheshwar@mayo.edu (J.B.)

**Keywords:** cytopathology, artificial intelligence, deep learning, rapid onsite evaluation

## Abstract

Artificial Intelligence (AI) has the potential to revolutionize cytopathology by enhancing diagnostic accuracy, efficiency, and accessibility. However, the implementation of AI in this field presents significant challenges and opportunities. This review paper explores the current landscape of AI applications in cytopathology, highlighting the critical challenges, including data quality and availability, algorithm development, integration and standardization, and clinical validation. We discuss challenges such as the limitation of only one optical section and z-stack scanning, the complexities associated with acquiring high-quality labeled data, the intricacies of developing robust and generalizable AI models, and the difficulties in integrating AI tools into existing laboratory workflows. The review also identifies substantial opportunities that AI brings to cytopathology. These include the potential for improved diagnostic accuracy through enhanced detection capabilities and consistent, reproducible results, which can reduce observer variability. AI-driven automation of routine tasks can significantly increase efficiency, allowing cytopathologists to focus on more complex analyses. Furthermore, AI can serve as a valuable educational tool, augmenting the training of cytopathologists and facilitating global health initiatives by making high-quality diagnostics accessible in resource-limited settings. The review underscores the importance of addressing these challenges to harness the full potential of AI in cytopathology, ultimately improving patient care and outcomes.

## 1. Introduction

The field of cytopathology, which involves the study and diagnosis of diseases at the cellular level, is on the cusp of a transformative era driven by advancements in artificial intelligence (AI). AI technologies, particularly machine learning and deep learning, have shown immense potential in revolutionizing medical diagnostics by providing enhanced accuracy, efficiency, and consistency [1]. These technologies can analyze vast amounts of data, recognize complex patterns, and make predictions with a level of precision that often surpasses traditional methods. In cytopathology, the integration of AI promises to address several long-standing challenges [2]. For instance, the variability in diagnostic interpretations due to human subjectivity can be mitigated through AI’s ability to provide consistent and reproducible results. Moreover, the labor-intensive nature of cytological assessments, which often involve meticulous examination of numerous cell samples, can be significantly alleviated by AI-powered automation [2,3]. This allows cytopathologists to allocate more time to complex cases and decision-making processes. Despite these promising advancements, the implementation of AI in cytopathology is not without its hurdles. One of the primary challenges is the need for high-quality, annotated data to train AI models to the point where it has prompted some scientists to resort to synthetically generated data [4]. The variability in sample preparation, staining techniques, and imaging modalities across different laboratories can complicate the development of robust and generalizable AI algorithms. Additionally, integrating AI tools into existing clinical workflows requires careful consideration of regulatory, ethical, and interoperability issues [5]. The necessity for thorough clinical validation to ensure that AI applications meet the stringent standards required for medical diagnostics further complicates their adoption.

Automated pre-screening solutions go back as far as the 1950s [6]. In recent years, artificial intelligence (AI) has made significant advancements in cytopathology, with the most well-known development being automated screening systems (Hologic ThinPrep Imager, BD FocalPoint for Surepath, PAPNET) for gynecologic cytopathology from the late 1990s. Around the same period, efforts were also devoted to developing classifier systems for other types of specimens, such as thyroid aspirates [7]. These early systems largely functioned by features manually engineered, extracted, and fed as continuous variables in a vector to a neural network. In more recent approaches, features are more commonly extracted through the process of training a CNN on annotated datasets.

The majority of AI research in cytopathology has been centered on gynecologic applications [8]. This review paper aims to provide a comprehensive overview of the current state of AI in non-gynecologic cytopathology, delineating the challenges that need to be addressed and the opportunities that lie ahead. We explore the advancements in AI technologies pertinent to cytopathology, examine the barriers to their widespread implementation, and highlight the potential benefits they offer in enhancing diagnostic capabilities and improving patient outcomes. By addressing these aspects, we hope to provide a balanced perspective on the integration of AI into cytopathology and its future trajectory in the realm of medical diagnostics. Additionally, we will emphasize that rapid onsite evaluation (ROSE) of cytology specimens, enhanced by AI, presents a transformative opportunity in the field. 

### Rapid Onsite Evaluation (ROSE) of Cytology Specimens

Rapid Onsite Evaluation (ROSE) is a service performed by cytopathology laboratories that provides assurance that the procedure adequately sampled the targeted lesion, captured enough samples to triage for ancillary testing, thereby avoiding repeat procedures [9], and may include a preliminary diagnosis from a qualified cytopathologist [10]. It is most commonly employed for thyroid [11], EUS (pancreas, lymph node, other GI tract) [12], EBUS (lung, lymph node, other respiratory tract) [13], and other percutaneous radiologically guided aspirate or tissue sampling (liver, kidney, bone, and soft tissue) [14]. We believe there is an opportunity to develop a tool that expands this service where ROSE may not be available dueto either resource constraints or a complete lack of availability. Diagnosis, sample triage, and ensuring enough material for ancillary testing are three very different tasks. AI-empowered ROSE holds the promise of delivering immediate feedback, ensuring better adequacy assessments, and streamlining workflows [15]. Incorporating AI into ROSE processes has the potential to revolutionize point-of-care diagnostics by significantly reducing turnaround times and improving accuracy. An AI-empowered solution is depicted in Figure 1.

## 2. Methods (Search Strategy)

Search terms (“cytology” OR “cytopathology”) AND (“artificial intelligence” OR “deep learning” OR “neural network”) were used to query the PubMed database over the last 5 years. The search returned 765 results as of 10 February 2024. Review articles covering general non-gynecologic cytopathology AI topics and articles about image-based deep learning (DL) algorithms were selected for this review. By narrowing the scope of our review to non-gynecologic cytology AI studies, we aimed to provide a more focused, detailed analysis of AI’s impact on this diverse and less standardized area. We refer readers to the study by Jiang et al. [16], which provides a comprehensive review of DL-based methods for cervical cytology screening. In total, 76 articles were reviewed, and 49 articles were determined to be in scope. The remainder were excluded upon further review of the text. The final selection of articles that characterized deep learning methods in the selected contexts was 21. We organized articles based on organs, including the thyroid, pancreatobiliary system (i.e., pancreas, bile duct, and gallbladder), and lungs. The PRISMA [17] flowchart illustrating the search methodology and exclusion criteria is shown in Figure 2, and the distribution of the included studies based on organs is shown in Figure 3 as a bar chart.

## 3. Results

Figure 3 highlights the relative scarcity of deep-learning-based studies performed on non-thyroid cytology specimens (EBUS samples for lung specimens and EUS samples for the pancreatobiliary specimens). Approximately 24% and 14% of studies fitting inclusion criteria were found to be focused on lung and pancreatobiliary cytology, respectively, with the remaining 62% focusing on thyroid FNAC. 

### 3.1. Thyroid

AI has most recently been applied to thyroid cytopathology in three cases: (1)in classifying benign versus malignant nodules, or other categories [18,19,20,21,22,23,24].(2)disambiguating equivocal diagnoses of atypia of undetermined significance/follicular lesions of undetermined significance (AUS-FLUS) [25].(3)location and segmentation of follicular cells [19,26].

The current gold standard for triage of these nodules is molecular testing, and to our knowledge, no current articles use molecular testing as an endpoint for this disambiguation. This may be due to the amount of data required to learn and balance multiple categories, as AUS-FLUS cases are relatively rare. The 2023 Bethesda System for Reporting Thyroid Cytopathology recommends limiting this diagnosis to no more than ≤7% of thyroid aspirates [27]. 

Many studies have been performed to predict malignancy in thyroid cytopathology. An early study by Varlatzidou et al. [28] proposed an AI approach by using LVQ43 (Learning Vector Quantizer Neural Networks). In this study, image features containing the size, shape, and texture of ~100 nuclei were first extracted from the WSI, and these features were applied to do the classification at the cellular level, followed by another classification at the individual patient level. The results indicated that using artificial neural networks (ANN) combined with image morphometry could potentially support thyroid lesion malignancy detection, especially in follicular neoplasms suspicious for malignancy and in Hurthle cell tumors.

To distinguish between papillary carcinoma thyroid (PTC) and non-papillary carcinoma thyroid (non-PTC), Sanyal et al. [18] trained a shallow CNN model on a limited number of microphotographs obtained at 10x and 40x magnification and evaluated the model on 174 microphotographs with a sensitivity of 90.48%. Similarly, Guan et al. [23] used VGG-16 and Inception-v3 to distinguish between PTC and benign thyroid nodules. Their VGG-16 model showed a better performance compared to the Inception-v3 model (accuracy: 97.66% vs. 92.75%). Elliott Range and Dov et al. [19] conducted a series of studies in 2020 and 2022 using thyroid fine-needle aspiration biopsy images. The authors utilized the VGG11 model pre-trained on ImageNet to predict the level of malignancy. The performance was compared with an experienced board-certified cytopathologist, and a follow-up histologic diagnosis study was performed with the same cytopathologist after a washout period of 117 days. The results indicated that the performance of the AI approach is comparable to the performance of an expert cytopathologist. This series of studies highlights the potential of AI techniques for future clinical use. In 2023, Hirokawa et al. [25] trained an EfficientNetV2-L model to analyze nodules that are diagnosed as follicular neoplasm (FN) or atypia of undetermined significance (AUS). This study pointed out that poorly differentiated thyroid carcinoma was challenging to distinguish from PTC, medullary, and FN. Zhu et al. [26] developed a hybrid method, including classification and segmentation branches for follicular cell segmentation of cytopathological thyroid WSI. The study cohort included 17 cases of PTC and 26 benign cases. They trained their model on 6900 cropped patches (1024 by 1024 pixels) from 13 WSIs and validated it on 30 WSIs. Their hybrid approach outperformed other popular deep learning benchmark architectures (mean accuracy: ~68% vs. ± 50% for FCN, Unet and DeeplabV3). However, the study includes several limitations, such as not using stratified split or k-fold cross-validation. Fragopoulos et al. [21] developed a radial basis NN method to predict benign and malignant lesions from liquid-based cytology samples. They extracted nuclear morphology features of 41324 nuclei from digitized WSIs of 447 patients. The model obtained 82.5% and 94.6% of sensitivity and specificity on the test set (50% of nuclei). The study limitations include not performing any benchmarking analyses for comparison and having no proper training and test split based on WSI. Additionally, using a random selection of cells could intuitively increase test performance. Gopinath et al. [22] in 2013 developed a computer-aided diagnostic system, including cell segmentation using threshold-based approach and cell classification based on the extracted morphological features to predict benign and malignant nodules using an Elman Neural Network (ENN). Their approach with ENN achieved a diagnostic accuracy of 93.3%. In 2015 [29], the same group combined the machine learning methods (k-NN, SVM, decision tree, and ENN) in several linear combinations and majority voting arrangements. None of the classifiers alone performed as well as the combined classifiers, all of which achieved the same diagnostic accuracy of 96.66%. 

Sensitivity and specificity varied between 95–100% and 90–100%, respectively. Savala et al. [24] in 2018 constructed an ANN to differentiate between follicular adenocarcinoma and follicular carcinoma. The authors selected multiple representative images of Giemsa-stained thyroid FNA smears photographed at 40x magnification. The ANN trained based on morphological features and achieved 100% accuracy on nine validation cases. It is very difficult to derive a conclusion from this study, as the dataset is small and no k-fold cross-validation was performed. Despite the fact that the accuracy of AI approaches in cytopathological diagnosis using thyroid fine-needle aspiration cytology images is comparable to human experts, these approaches have mostly been limited to distinguishing between two lesions (e.g., benign and malignant) and evaluated on limited datasets. These limitations could potentially result in overfitting and low generalizability to external cohorts and might suffer from out-of-distribution errors. 

The performance and details of the studies are summarized in Table 1.

### 3.2. Pancreatobiliary System

Pancreatobiliary cancers are known for their aggressive nature and often poor prognosis due to late-stage diagnosis [30,31]. The key types include pancreatic cancer, cholangiocarcinoma (bile duct cancer), and gallbladder cancer. Momeni-Boroujeni et al. [32] used a feed-forward multilayer perceptron to classify microphotographs of thinprep cell clusters. The dataset used in this study is limited (i.e., 277 images of cell clusters, including 118 benign, 74 malignant, and 85 atypical) and was split with a 7:3 ratio for training and test purposes. They reported 100% accuracy for malignancy prediction and 77% accuracy for atypical cases. The authors concluded that their model can be used to distinguish benign from malignant pancreatic cytology, as well as to distinguish atypical cases. Lin et al. [33] performed a prospective study to develop an image patch classifier designed to separate images with and without cancer cells with a ResNet101V2 neural network. DiffQuik-stained EUS-FNA slides were photographed at 20x objective magnification on BX53F or IX73 microscopes (Olympus Corporation, Tokyo, Japan). Individual images were classified by a pathologist, and malignant/suspicious were classified as positive and further annotated. Images were normalized for stain color and 256-pixel square tiles with 50% overlap were extracted. These tiles were divided into positive and negative, and a final training and internal validation dataset were constructed (3642 tiles from 367 images, and 916 tiles from 100 images, respectively). After training, an external test dataset consisting of 693 images, classified by a pathologist, was used to test the algorithm. The model’s performance on the external dataset was 88.7% accuracy with a sensitivity of 78.0%, specificity of 90.6%, PPV of 60.7%, and NPV of 95.7%, demonstrating the potential usefulness of off-the-shelf image classification algorithms as part of an on-site evaluation process for pancreas EUS-FNA. However, the paper did not address the speed of inference on new data. Zhang et al. [34] developed two models in the ensemble, a UNet-based CNN with ResNet101 encoder/decoder to classify cell clusters at the pixel level as pancreatic cancer positive versus negative. The datasets consisted of 5345 resized images of Giemsa-stained smears captured by photomicrograph at 40x objective magnification on an Olympus DP73 camera (Olympus Corporation, Tokyo, Japan) mounted on an Olympus BX40 microscope. Data augmentation techniques of random rotation, horizontal and vertical flips, and brightness and hue transformations, were also performed on the training images. The trained algorithm was measured on the segmentation task on three levels: across all cell clusters, per image adequacy threshold, and per patient adequacy threshold calculated separately on their internal and external datasets. Classification accuracy for pancreatic cancer positive versus negative was tested on internal, external, and separate human–machine competition datasets. The latter were scored against human performance. On Human–Machine Competition Dataset 1, the algorithm achieved a sensitivity of 98.7% (95% CI 96.9–100.0), specificity of 93.0% (95% CI 90.5–95.4), PPV of 83.5% (95% CI 78.1–88.9), NPV of 99.5% (95% CI 98.8–100.0), and diagnostic accuracy of 94.5% (95% CI 93.3–95.5). This showed higher diagnostic accuracy, significantly higher sensitivity and NPV, and comparable PPV and specificity to cytopathologists in this study. Kruita et al. [35] investigated the diagnostic ability of AI to differentiate malignant from benign pancreatic cystic lesions. They trained an ANN with features extracted from 85 patients, including carcinoembryonic antigen in the cyst fluid, clinical data (e.g., sex, cyst location, type of cyst), and cytology, and performed 5-fold cross-validation to assess the performance of their model. The diagnostic ability of their AI model was as follows: 95.7% sensitivity and 91.9% specificity. The current AI studies on pancreatobiliary in the literature were evaluated on specific targets, such as image tiles, including cell clusters. The prediction results are not based on WSI, and no strategies were presented for patient WSI-level predictions. The AI models are still in their infancy stages and further work needs to be performed to address these limitations. 

The performance and details of the studies are summarized in Table 2.

### 3.3. Lung

Lung cancer is the second most common cancer in both men and women in the U.S.; however, it is the leading cause of cancer-related deaths. In 2023, it is estimated that there will be about 238,340 new cases of lung cancer and approximately 127,070 deaths due to lung cancer in the United States [36,37,38]. The early detection of lung cancer is usually performed with computer tomography (CT). If a suspected lesion is found by CT screening, a subsequent cytological diagnosis is conducted using bronchoscopic biopsy. Several studies [39,40,41,42,43,44,45] have utilized DL methods to assess cytology specimens of the lung. Lin et al. [42] used a hybrid classification and segmentation strategy to develop a deep learning-based system for ROSE on a prospective study dataset. The dataset consisted of microphotographs at 10x, 20x, and 40x objective magnification. The dataset of 97 patients was split into 70% for training and 30% for validation. In total, 499 preprocessed images, which were divided into 7486 image patches, were used to train a ResNet101 algorithm for identifying patches with malignant cells. The algorithm performed with a classification accuracy of 98.8% at the patch level, 95.5% at the image level, and 92.9% at the patient level. An HRNet semantic segmentation algorithm was used to segment the malignant cell area on the positive patches, achieving a mIoU of 89.1%. Teramoto et al. [39,40] published two studies focusing on classification strategies and data augmentation techniques for diagnosing cancers from Pap-stained liquid-based cytology images. The first study, using a VGG-16-based model and additional data augmentation techniques, improved accuracy to 87.0% for distinguishing between benign and malignant images. The second study further enhanced accuracy by adding 10,000 synthetic images generated by a GAN, raising accuracy by 4.3% to 85.3%. Gonzalez et al. [43] trained three separate Inception V3 algorithms to differentiate small cell lung carcinoma (SCLC) from large cell neuroendocrine carcinoma (LCNEC) using Pap-stained smears, DiffQuik-stained smears, and H&E-stained tissue biopsies. The dataset consisted of 40 cases, 114 slides, and 464,378 image patches. Each algorithm achieved high accuracy: 85.7% for SCLC and 100% for LCNEC using Pap-stained smears, while both the DiffQuik and H&E algorithms reached 87.5% and 100% accuracy for SCLC and LCNEC, respectively. The authors noted that their study is limited due to the small size of their dataset.

Considering the potential for bias in cytopathology, data used for ROSE are markedly different from archival cytopathology data. In a point-of-care setting, the slide will be very fresh (liquid on slide, limited time for stain to fade or bleed) and likely only in focus at no more than 200x due to specimens being uncoverslipped. Study design may need to take this into account with at least a partially prospective design. Furthermore, in Ishii et al., 2022 [44], a machine learning-based approach was designed to predict gene alterations in cytologic samples from primary lung cancer, providing a cost-effective and efficient alternative to traditional gene testing. The model demonstrated strong predictive performance for EGFR and KRAS mutations, with accuracy nearing 95%, while predictions for ALK-fusion and cases without mutations showed moderate accuracy at approximately 75% and 80%, respectively. Additionally, the model identified key cytological characteristics, including a unique feature associated with EGFR mutations, highlighting its potential to support precision medicine in lung cancer care. It is worth noting, anecdotally, in cytopathology practices, that specimens referred to as FNA may also include touch imprint preparations from core needle biopsies. We believe it is important to include these in training and test data to address the potential for bias.

The performance and details of the studies are summarized in Table 3.

## 4. Discussion

AI has emerged as a transformative force in cytopathology, offering the potential to enhance diagnostic accuracy, streamline workflows, and expand access to high-quality care. In this review, we presented the multifaceted role of AI in cytopathology, emphasizing critical challenges and opportunities that shape its adoption and impact. Key discussion points include quality control, data augmentation, model performance, manual annotation challenges, human–machine collaboration, clinical integration, and future trends. By addressing technical hurdles such as z-stack scanning, artifact detection, and computational infrastructure, alongside innovative solutions like weakly supervised learning and hybrid models, this paper outlines a comprehensive pathway to integrating AI into cytopathology. It underscores the necessity of collaborative efforts to ensure AI systems are accurate, user-friendly, and incorporated into clinical workflows, ultimately advancing patient outcomes and the field’s evolution.

### 4.1. Data Augmentation, Model Performance, and Reproducibility

In deep learning, data augmentation is a key technique that bolsters model robustness by introducing variability into the training data, thereby mitigating bias and improving overall performance. This is especially crucial in biomedical imaging, where variations in scanner hardware, software, staining methods, and fixatives can cause data inconsistencies, potentially skewing deep learning models. Techniques such as color adjustment—through normalization and manipulation of hue, saturation, and contrast—are commonly used to address these discrepancies. Other methods, including noise filtering, edge enhancement, image rotation, random cropping, and mirroring, further enrich the training dataset’s diversity. Additionally, generative adversarial networks (GANs) have become a powerful tool for creating synthetic data, which, when added to training sets, significantly enhances model performance. These strategies not only help reduce bias from variable data characteristics but also expand the dataset, thereby improving the overall accuracy and robustness of deep learning models in biomedical image analysis.

In cytology, the reporting of performance metrics has been inconsistent due to the relatively recent application of deep learning and the variety of clinical tasks to which it is applied. While diagnostic accuracy is the most frequently reported metric, other important measures, such as sensitivity (recall) and specificity, are less commonly reported, and even fewer studies include positive and negative predictive values, AUC (Area Under the Curve), or precision and F1-scores. Task-specific metrics like mean intersection over union (mIoU) are typically used for segmentation tasks, underscoring the need to choose appropriate metrics based on the specific clinical objective, whether for screening or diagnosis. Relying solely on accuracy can be misleading, as it fails to differentiate between types of errors, such as false positives and false negatives, which have distinct clinical implications. A thorough analysis of misclassified cases is, therefore, crucial for refining models to meet the nuanced needs of clinical applications and diverse patient populations.

Cytology deep learning research often focuses on specialized tasks such as segmentation and classification. Segmentation, as a foundational step, enables higher-order analyses like morphometric evaluations. Neural networks trained to segment features like nuclei can provide valuable insights for downstream tasks, with parameters like nuclear size, shape, variation, and the nuclear-to-cytoplasmic ratio serving as key indicators of malignancy. Extracting these morphometric features is a critical step in generating inputs for machine learning models, enhancing their capacity to detect malignancies in cytology.

Classification also plays a vital role, particularly in WSI, where tile-based classification helps identify regions of interest for screening. Classification tasks vary from evaluating specimen adequacy to distinguishing malignant from non-malignant cells, providing multiclass cytological diagnoses, or resolving ambiguous diagnostic categories. More advanced applications include predicting molecular mutations and pushing the boundaries of deep learning in clinical cytology.

Hybrid models that combine segmentation and classification offer improved efficiency and computational savings. For instance, Zhu et al. [26]. proposed a method that integrates a custom ResNet101-based classification step, followed by segmentation using atrous spatial pyramid pooling (ASPP). This hybrid approach maintained similar statistical performance to traditional segmentation methods like UNet while significantly reducing computation time. These types of models demonstrate the potential for combining tasks, leading to more efficient and scalable deep learning systems for cytology applications.

### 4.2. Quality Control and Assurance

In cytology, quality control and assurance are pivotal for ensuring accurate and reliable results, especially in the context of WSI. Scanning thicker cytopathological preparations, typically 40–50 microns compared to thinner histological samples, introduces unique challenges. These include the need for z-axis scanning, which significantly increases the time and complexity of the process, as well as larger file sizes that demand more storage and higher bandwidth. To accommodate these demands, a robust IT infrastructure is essential.

Additionally, WSI optics face a trade-off between resolution and field of view. Lower magnification offers a broader view but sacrifices detail, while higher magnification enhances detail at the cost of a narrower field of view. A promising solution is Fourier Ptychographic Microscopy (FPM) [46], which increases the depth of field without moving parts. By capturing images at various light source angles and using computational techniques to combine them into a single, in-focus image, FPM addresses both z-axis and dimensionality challenges. Recent advancements have focused on accelerating image acquisition and processing through LED multiplexing and deep learning, reducing the time required and increasing the efficiency of cytopathological scanning and quality control [46].

Artifact detection, modification, and removal are equally critical for accurate computational analysis and clinical diagnosis. Preparation artifacts like broken coverslips, media bubbles, or focus issues, as well as electronically generated artifacts, such as image compression or WSI tile stitching, can obscure key sample features, potentially compromising the entire scan.

To address these issues, several innovative tools have been developed. Janowczyk et al. [47] created HistoQC, an open-source software that detects artifacts, generates masks, and offers metrics for identifying outliers and batch effects. This tool is particularly useful in clinical workflows, enabling operators to identify slides that require reimaging or reprocessing before diagnosis. In addition, Jiang et al. [48] applied a generative adversarial network (GAN) based on Pix2Pix to remove pen markings from histology slides, further improving image clarity for diagnostic purposes. These advancements highlight the growing impact of computational tools in enhancing the quality and usability of WSI data, leading to more accurate and reliable diagnostic outcomes.

### 4.3. Tedious and Laborious Manual Annotation

Annotation in cytology is a labor-intensive task that demands significant expertise from cytologists, often pathologists or cytotechnologists. Due to the scarcity of cytologist time and the high costs associated with their expertise, coupled with data privacy, ownership concerns, and the expenses of data generation and storage, high-quality cytopathology datasets are rare. Most publicly available datasets are from Pap tests, with fewer datasets available for other areas like breast, stomach, or oral cytology. Moreover, it is uncommon to find cytology datasets in Whole Slide Imaging (WSI) format, as most data exists in the form of photomicrographs or single-cell images.

To overcome the challenges of exhaustive manual annotation, weakly supervised methods like Multiple Instance Learning (MIL) offer an efficient alternative for cytopathology applications. MIL allows for the classification of “bags” of features rather than requiring detailed annotations for entire WSIs. For instance, a bag containing lymphocytes, macrophages, and germinal center features could indicate a negative classification for lymph node samples, while a bag with malignant cells would likely lead to a positive classification. This approach significantly reduces the time and cost of manual annotation while still producing reliable results. For example, Campanella et al. [49] achieved remarkable clinical-grade outcomes with an AUC of 0.98 on histopathology slides using case-level diagnoses as labels, bypassing the need for extensive data curation. This demonstrates the potential of MIL to streamline annotation processes in cytopathology without sacrificing diagnostic accuracy.

When developing AI algorithms for cytopathology, it is crucial to balance accuracy and speed, considering the intended use case. For instance, algorithms designed to augment Rapid Onsite Evaluation (ROSE) must prioritize speed, even if it means sacrificing some accuracy. In contrast, algorithms meant to resolve equivocal diagnosis, which can guide therapeutic decisions or inform costly ancillary tests, must achieve higher accuracy. The volume of data processed also affects algorithm efficiency, as high volumes can quickly accumulate GPU time, influencing economic decisions, such as whether to invest in GPU hardware or rent computing resources. Additionally, the operator’s expertise must be considered; an algorithm designed for use by a skilled cytotechnologist will have different requirements than one aimed at extending near-human performance to another specialist.

### 4.4. Human–Machine Collaboration in Cytopathology

While many articles discuss human–machine collaboration in cytopathology, few address the key challenge of developing AI tools that are not only accurate and generalizable but also fast, transparent, and user-friendly. Tools must inspire trust and be easy to use, as even highly capable algorithms may fail to gain adoption if their interfaces are cumbersome. One example of an impressive tool is an augmented reality microscope by Chen et al. [50], which integrates features such as cell and mitosis counting, stain quantification, and microorganism detection. The system overlays heatmaps to indicate uncertainty, outlines tumor grading predictions, and provides notifications. A similar tool could benefit cytopathology, particularly for those who prefer using traditional glass slides, allowing them to leverage AI’s efficiency while maintaining familiar workflows.

However, the increasing reliance on AI in pathology raises concerns about deskilling. There is a fear that experts may become overly dependent on AI, potentially losing the fine skills necessary to make nuanced diagnostic decisions. This risk is particularly relevant for new trainees, who may fail to develop essential diagnostic abilities if they rely too heavily on AI during their training. Balancing AI use with maintaining human expertise will be critical as AI continues to influence clinical practice. Moreover, reinforcement learning could be explored as a way to evolve AI algorithms in response to real-world practices, ensuring continuous improvement while preventing issues like algorithmic drift, as discussed in the FDA’s Software as a Medical Device (SaMD) white paper on quality assurance and improvement programs [51,52].

### 4.5. Integration into Cytopathology Clinical Workflow

Integrating digital cytopathology AI tools into existing laboratory workflows presents several challenges, despite their potential to enhance diagnostic accuracy and efficiency. One significant hurdle is the limited availability and high cost of digital cytopathology scanners, which are essential for capturing high-resolution images of cytological slides. These scanners must be compatible with diverse specimen types and staining methods, which vary widely between labs, potentially necessitating workflow modifications. Furthermore, the implementation of AI tools requires robust computational infrastructure and trained personnel to manage and interpret results, which may strain resources in laboratories already operating under tight budgets or staffing constraints. Another challenge lies in ensuring integration with existing laboratory information systems (LIS), requiring complex customizations to facilitate data flow and maintain compliance with regulatory standards. Additionally, pathologists may need time and training to adapt to AI-assisted diagnostics, balancing traditional methodologies with digital tools while ensuring diagnostic accuracy and reproducibility. These barriers underscore the need for strategic planning and collaboration among stakeholders to achieve successful integration.

### 4.6. Future Trends and Directions

The future of AI in cytology is poised for transformative advancements, driven by innovations in deep learning, computational efficiency, and data accessibility. One of the key trends is the development of more sophisticated hybrid models that integrate multiple tasks, such as segmentation, classification, and morphometric analysis, into a single, streamlined pipeline. This not only improves diagnostic accuracy but also reduces the time and resources needed for processing. The use of generative adversarial networks (GANs) for data augmentation and synthetic data generation will likely continue to grow, addressing the issue of limited high-quality cytology datasets. Additionally, AI tools designed to collaborate with human experts are becoming increasingly user-friendly, with augmented reality and integrated systems that enhance cytologist workflows while reducing cognitive load for routine or error-prone tasks. Another emerging trend is the shift toward real-time AI assistance in clinical settings, such as during Rapid Onsite Evaluation (ROSE), where speed and accuracy are paramount. Finally, as regulatory frameworks for AI-driven medical tools evolve, there will be a greater emphasis on transparency, generalizability, and trustworthiness, ensuring that AI systems are safe, effective, and widely adopted in clinical practice. These advancements will continue to push the boundaries of AI in cytology, improving diagnostic precision, and expanding access to high-quality care.

## 5. Conclusions

The integration of AI into cytology holds immense potential to transform diagnostic practices by improving accuracy, efficiency, and accessibility. AI-driven technologies, from image analysis to classification and automation, offer solutions to the challenges of manual cytology, such as human error, variability, and the labor-intensive nature of screening. Despite significant advancements, the field faces challenges in data availability, standardization, and model interpretability. Overcoming these barriers will require collaborative efforts to create more diverse, high-quality datasets and to develop explainable AI models that inspire trust among clinicians. As AI continues to evolve, it is poised to not only enhance the diagnostic capabilities of cytologists but also expand access to high-quality cytology services, particularly in underserved areas. The future of AI in cytology promises to deliver more accurate, scalable, and equitable diagnostic tools, ultimately improving patient outcomes across the globe.

## Figures and Tables

**Figure 1 bioengineering-12-00176-f001:**
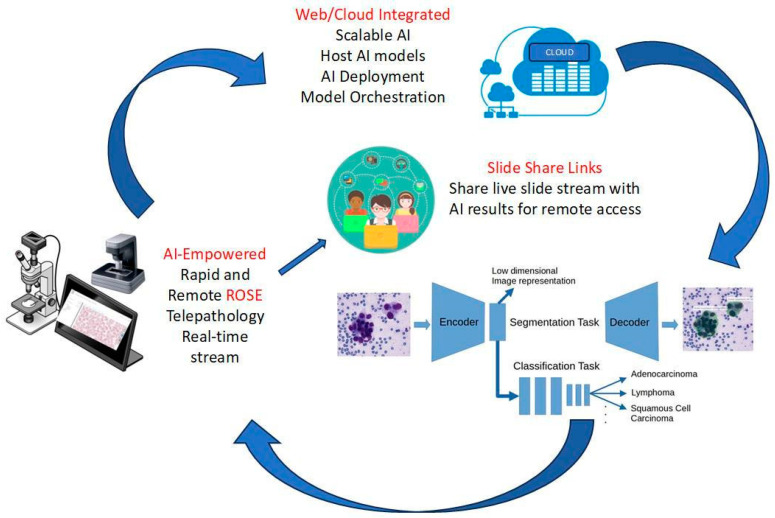
A depicted framework of AI-empowered remote and rapid onsite evaluation of cytology specimens.

**Figure 2 bioengineering-12-00176-f002:**
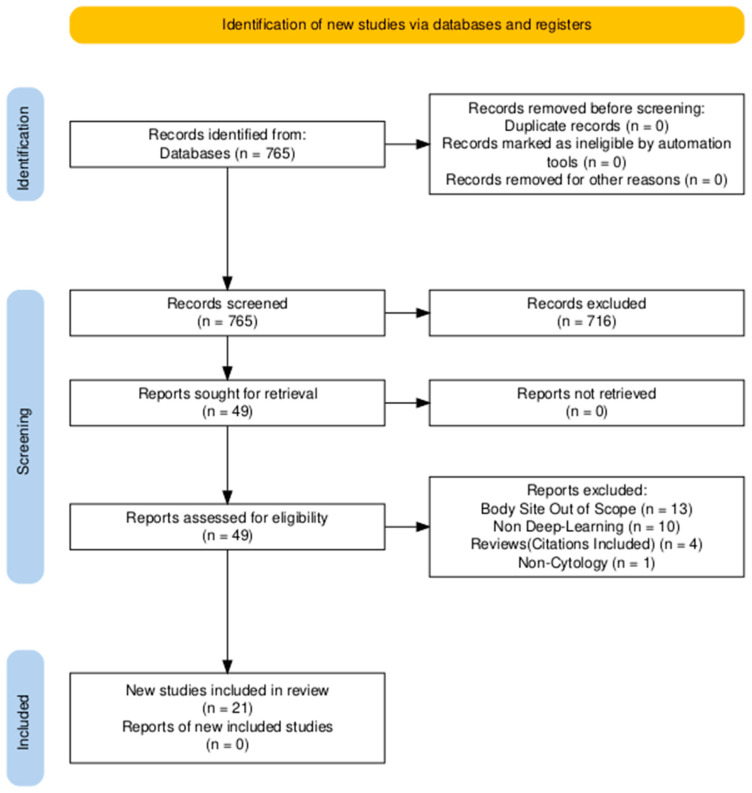
PRISMA flowchart [17] illustrating the search methodology and exclusion criteria.

**Figure 3 bioengineering-12-00176-f003:**
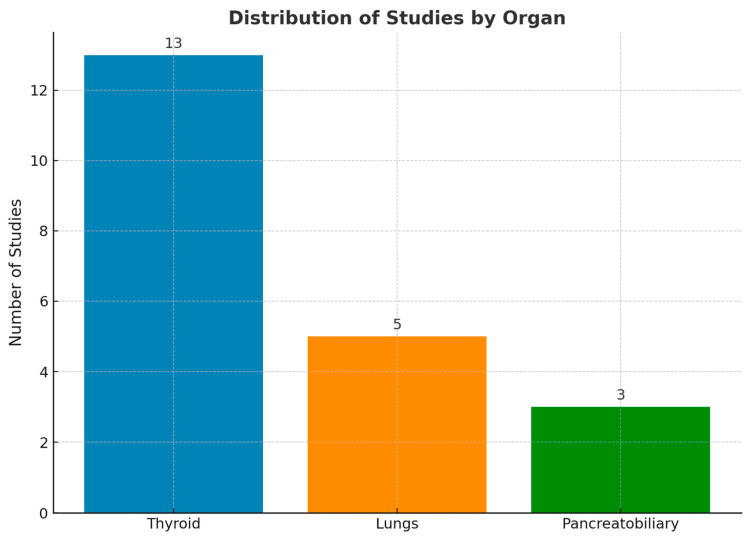
Bar chart illustrating the distribution of included studies based on organ.

**Table 1 bioengineering-12-00176-t001:** Characteristics of AI Models for Thyroid Cytology Classification.

Study	Objective	Dataset Total/Test	Method	Performance Metrics
Hirokawa et al., 2023 [25]	AUS classification	393 nodule, 148,395 patch, 9782 images	EfficientNetV2-L	Sens: 94.7 Spec: 14.4 PPV: 56.3 NPV: 66.7 PR AUC: >0.95
Dov et al, 2021 [20]	TBSRTC category	254/109	VGG11, ImageNet	AUC: 0.931
Zhu et al., 2021 [26]	Semantic Segmentation vs. Patch Classification	6900/2400	Enhanced ASPP/ Integrated Classifier (ResNet 101 basis)	Segmentation: AUC: 99.50 Classification: Acc: 99.3
Fragopoulos et al., 2020 [21]	Benign vs. Malignant	447/223	ANN	Nucleus Level-Acc: 86.94 Sens: 81.37 Spec: 90.01 PPV: 81.74 NPV: 83.78 Patient Level-Acc: 95.07 Sens: 95.00 Spec: 95.10 PPV 91.57 NPV 97.14
Elliott Range et al., 2020 [19]	TBSRTC category	918/109	VGG11, ImageNet	Sens 92.0% Spec 90.5% AUC 0.932
Zhu et al., 2019 [26]	AUS vs Malignant	467	DNN NOS	Sens: 87.91 Spec: 85.15 AUC: 0.891 Acc: 87.15
Guan et al, 2019 [23]	PTC vs benign nodules	887/128	DCNN VGG-16	Acc 97.66% Sens 100% Spec 94.91% PPV: 95.83% NPV: 100%
Savala et al., 2018 [24]	Follicular adenoma vs. carcinoma	57/9	ANN	Sens: 100 Spec: 100 AUC: 1.00
Sanyal et al, 2018 [18]	PTC vs. Non-PTC	418/48	CNN	Sens: 90.48 Spec: 83.33 PPV: 63.33 NPV: 96.49 Acc: 85.06
Gopinath et al., 2015 [29]	Benign vs. Malignant	110/30	SVM, Elman ENN	Sens: 95 Spec: 100 Acc: 96.66
Gopinath et al., 2013 [22]	Benign vs. Malignant	110/30	ENN/SVM	Sens: 100 Spec: 80 Acc:93.33
Varlatzidou et al., 2011 [28]	Benign vs. Malignant	335/50%	LVQ43 Learning Vector Quantizer NN	Sens: 91.51 Spec: 92.43 PPV: 74.26 NPV: 97.85

AUC: Area Under Curve; Acc: Accuracy; Sens: Sensitivity; Spec: Specificity; CNN: Convolutional Neural Network; ENN: Elman Neural Network; TBSRTC: The Bethesda System for Reporting Thyroid Cytopathology.

**Table 2 bioengineering-12-00176-t002:** Characteristics of AI Models for Pancreatobiliary Cytology Classification.

Study	Objective	Dataset Train/Test	Method	Performance Metrics
Zhang et al., 2022 [34]	Segmentation/Pancreatic Ca vs. Non-Ca	Patients: 194 Total: 5345 images Train: 2166 images, 66 patients Val: 695 images, 16 patients Test: 1162 images, 27 patients (internal); 1322 images, 85 patients (external)	UNet-based CNN	Segmentation: Acc: 0.964 F-Sc: 0.929 Prec: 0.927 Rec: 0.931 Diagnosis: Acc: 0.945 Sens: 0.987 Spec: 0.930 PPV: 0.835 NPV: 0.995
Kruita et al., 2019 [35]	Benign vs. Malignant	Patients: 85 Train: 68 Test:	multi-hidden layer of neural network	AUC: 0.966 Sens:95.7% Spec:91.9%
Momeni-Boroujeni et al., 2017 [32]	Benign vs. Malignant	Total Dataset: 75 cases Train N: 70%Test N: 30%	MLP (feed-forward)	Sens: 80% Spec: 75% Acc: 100.0 AUC: 0.917

AUC: Area Under Curve; Acc: Accuracy; Sens: Sensitivity; Spec: Specificity; CNN: Convolutional Neural Network.

**Table 3 bioengineering-12-00176-t003:** Characteristics of AI Models for Lung Cytology Classification.

Study	Objective	Dataset Total/Test	Method	Performance Metrics
Ishii et al., 2022 [44]	ALK vs. EGFR vs. KRAS vs. None EP: Molecular Gene Alteration	Total: 40 cases, 114 slides, 464,378 patches, 145,468 H&E patches Train N: 20 cases Test N: 20 cases	MobileNet-V2 Transfer Learning	Sens: Patch level: 0.688 (ALK); 0.933 (EGFR); 0.942 (KRAS); 0.450 (None) Spec: Patch level: 0.778 (ALK); 0.986 (EGFR); 0.948 (KRAS); 0.961 (None) Acc: Patch level: 0.760 (ALK); 0.969 (EGFR); 0.947 (KRAS); 0.809 (None) Prec: Patch level: 0.432 (ALK); 0.968 (EGFR); 0.811 (KRAS); 0.830 (None) F-Sc: Patch level: 0.530 (ALK); 0.950 (EGFR); 0.871 (KRAS); 0.584 (None)
Lin et al., 2021 [42]	Semantic Segmentation/ Diagnosis: Benign vs. Malignant	Total: 499 images, 7486 patches Train N:70% Val N: 15% Test N:15%	HRNet/ResNet101	Segmentation: AUC: 89.2 Diagnosis PL: Sens: 98.8 Spec: 98.9 PPV: 99.1 NPV: 98.3 Acc: 98.8 Diagnosis IL: Sens: 98.2 Spec: 77.8 PPV: 96.6 NPV: 87.5 Acc: 95.5
Gonzalez et al., 2020 [43]	SCLC vs. LCNEC	40 cases, 114 slides, 464,378 total patches, 59,072 Pap patches	Inception V3	Sens: 1 Spec: 0.875 AUC: 0.875 Acc: 87.5 (SCLC); 100.0 (LCNEC)
Teramoto et al., 2020 [40]	Benign vs. Malignant	60 cases, 511 images, 793 patches	Modified VGG-16	Sens: 85.4 Spec: 85.3 Acc: 0.853
Teramoto et al., 2019 [39]	Benign vs. Malignant	47 cases/417 images/621 patches	Modified VGG-16, ImageNet Transfer Learning	Sens: 89.3 Spec: 83.3 AUC: Patch level: 0.872; Case level: 0.932

AUC: Area Under Curve; Acc: Accuracy; Sens: Sensitivity; Spec: Specificity.

## Data Availability

No new data were created or analyzed in this study.

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
