# Peer review of "Challenges and Opportunities in Cytopathology Artificial Intelligence"

_bioengineering, 2025, doi:10.3390/bioengineering12020176_

Round 1
Reviewer 1 Report
Comments and Suggestions for Authors
The paper is well written and provides valuable insights to be considered in the field of artificial intelligence application to diagnostic cytopathology.
The review highlights both the challenges and opportunities that AI introduces to cytopathology, emphasizing critical tasks such as acquiring quality-labeled data, developing robust algorithms, ensuring standardization and clinical validation. The Authors trace the evolution of AI in cytopathology, from its early approaches to the most recent technologies. therefore, the manuscript remain relevant not only despite the numerous reviews already published in the field of cytopathology but also as source of updates . The paper, although limited to the lung, thyroid and hepatobiliary system, provides an update on this topic. It focused on challenges points and offers suggestions on technical aspects that can influence the processes and performance of deep learning models, which are well reported throughout the discussion. The conclusions are consistent with the the nature of the work. Regarding to the application of AI for rapid on site evaluation, Authors could update their work with the most recent data reported in the literature in this field
The tables and figures appear easy to interpret
Author Response
Reviewer Comment no.1:
The paper is well written and provides valuable insights to be considered in the field of artificial intelligence application to diagnostic cytopathology.
The review highlights both the challenges and opportunities that AI introduces to cytopathology, emphasizing critical tasks such as acquiring quality-labeled data, developing robust algorithms, ensuring standardization and clinical validation. The Authors trace the evolution of AI in cytopathology, from its early approaches to the most recent technologies. therefore, the manuscript remain relevant not only despite the numerous reviews already published in the field of cytopathology but also as source of updates. The paper, although limited to the lung, thyroid and hepatobiliary system, provides an update on this topic. It focused on challenges points and offers suggestions on technical aspects that can influence the processes and performance of deep learning models, which are well reported throughout the discussion. The conclusions are consistent with the the nature of the work. Regarding to the application of AI for rapid on site evaluation, Authors could update their work with the most recent data reported in the literature in this field
Response to Reviewer Comment no.1:
Thank you for your insightful and constructive comments. In addition, changes have been made to the manuscript regarding the section on rapid on-site evaluation (ROSE) in lines 88-92 of manuscript and section 4.3. in the discussion.
Reviewer 2 Report
Comments and Suggestions for Authors
This manuscript is a systematic review of recent and possible applications of artificial intelligence (AI) in cytopathology, in particular for non-gynecologic use-cases. Some issues might be considered:
1. The focus on non-gynecologic applications might be included in the manuscript title.
2. For the flowchart in Figure 2, the exclusions from n=76 to n=49 should be reflected in the box next to n=76.
3. In Table 1, "Bening" might be "Benign".
4. The studies covered appear to generally take raw images as input. It might be commented as to whether (prior) research has attempted classification on features extracted from images.
Author Response
Comments from reviewer 2
This manuscript is a systematic review of recent and possible applications of artificial intelligence (AI) in cytopathology, in particular for non-gynecologic use-cases. Some issues might be considered:
- The focus on non-gynecologic applications might be included in the manuscript title.
- For the flowchart in Figure 2, the exclusions from n=76 to n=49 should be reflected in the box next to n=76.
- In Table 1, "Bening" might be "Benign".
- The studies covered appear to generally take raw images as input. It might be commented as to whether (prior) research has attempted classification on features extracted from images.
Response to reviewer 2
Thank you for your insightful and constructive comments. Kindly find, changes made to reflect on them respectively:
- The introduction (lines 64-67) and methods section (lines 101-105) of the article highlights the rationale behind the exclusion of gynecological cytopathology articles/studies and directs the readers to a review of the ample research that has been undertaken over the years. We aim that the neutrality of the title will further highlight the lack of research being performed in other crucial areas of cytopathology as those included.
- Changes have been made to the PRISMA flowchart to correct the arithmetic error. There was a data entry issue in the shiny app (referenced in the caption of the PRISMA flowchart) that resulted in this discrepancy.
- Changes and spelling revisions have been made to the tables.
- The review focuses on studies that use deep learning algorithms with self-feature extraction abilities from raw images. There are 10 studies that focus on extracted features in a classical machine learning context, however, these studies were excluded, and this was highlighted in the PRISMA flowchart under “Non Deep-Learning” category.
Reviewer 3 Report
Comments and Suggestions for Authors
To the authors
The authors conducted a systematic review of artificial intelligence related to cytopathology. The authors searched hundreds of scientific papers from PubMed DB within the last 5 years, and segemented previous studies into three categories based on the organs. They reviewed the cases, algorithms, performances, and limitations of each reference. As a review article, selection process of the studies seems to be relevant. However, the authors should address the issues commented as below.
Major points
1. Review articles should cover diverse accumulated reference, considering the passage of time and adavancement in technology. When the authors develop the logics of the study, almost every point of information would need appropriate references. However, the authors cited only one reference by Trisolini, R. et al. from Reference section in Introduction section, and the other sections, especially Discussion section, also have scarce references, relatively. It would be the most critical issue for the authors to address, and it might require a modification of the entire manuscript.
2. The structure of Introduction section and the intention of 'Rapid Onsite Evaluation (ROSE) of Cytology Specimens' subsection are confusing. The caption text of Fig 1 should also show sufficient and detailed contents. Although the authors wrapped up Introduction and suggest the purpose of the study, what would be this subsection for? I recommend the authors reconstruct the Introduction section, or move the subsection to Methods section.
3. Figures in this study lack of relevant explation, and even Fig 2, 3 are not commented in the main text. If the figures are essential for this study, the authors should make relevant linkages between the main text and the figures, and manifest them to the readers.
4. The last paragraph of 3.1 Thyroid section occupies a full page, and the it has simply enumerated the number of data, adopted algorithms, performance, and limitation of each reference. This might be just another information of Table 1, and lack readability. I recommend the authors take a closer look into the references, and categorize them according to the study design and the types of neural networks. It seems that the same work is need in the other sections as well.
5. What are the point of discussion in Discussion section? Is is focused on the limitations of the present technologies and solution suggestions? Each of the six subsections have been written in different directions, but it is hard to determine and distinguish the intent of the writing, especially compared to Result section. If the authors have tried to illustrate the limitations of the state-of-the-art AI algorithms of previous studies, which are entirely dealt with in Results section, the structure of the manuscript should be modified. I recommend the authors reconstruct Result and Discussion sections for efficient communication of the authors' intention. There is no need for the structure of a review article to be identical to that of an original research article.
Minor points
1. Please check and correct typos. For example, a vertical bar next to 'Abstract' of Line 11 seems to be a typo. Feb 10, 2024.Review of Line 90 also should be checked. I'll not designate the others further.
2. Please standardize the notation of references. For example, from Line 52 to 61 and from Line 74 to 79 show inconsistent notation of the references, and I couldn't find them from Reference section. If the references are essential in this study, please add them to the Reference section. I'll not designate the other citations.
Author Response
Comments from reviewer 3
The authors conducted a systematic review of artificial intelligence related to cytopathology. The authors searched hundreds of scientific papers from PubMed DB within the last 5 years, and segemented previous studies into three categories based on the organs. They reviewed the cases, algorithms, performances, and limitations of each reference. As a review article, selection process of the studies seems to be relevant. However, the authors should address the issues commented as below.
Major points
- Review articles should cover diverse accumulated reference, considering the passage of time and adavancement in technology. When the authors develop the logics of the study, almost every point of information would need appropriate references. However, the authors cited only one reference by Trisolini, R. et al. from Reference section in Introduction section, and the other sections, especially Discussion section, also have scarce references, relatively. It would be the most critical issue for the authors to address, and it might require a modification of the entire manuscript.
- The structure of Introduction section and the intention of 'Rapid Onsite Evaluation (ROSE) of Cytology Specimens' subsection are confusing. The caption text of Fig 1 should also show sufficient and detailed contents. Although the authors wrapped up Introduction and suggest the purpose of the study, what would be this subsection for? I recommend the authors reconstruct the Introduction section, or move the subsection to Methods section.
- Figures in this study lack of relevant explation, and even Fig 2, 3 are not commented in the main text. If the figures are essential for this study, the authors should make relevant linkages between the main text and the figures, and manifest them to the readers.
- The last paragraph of 3.1 Thyroid section occupies a full page, and the it has simply enumerated the number of data, adopted algorithms, performance, and limitation of each reference. This might be just another information of Table 1, and lack readability. I recommend the authors take a closer look into the references, and categorize them according to the study design and the types of neural networks. It seems that the same work is need in the other sections as well.
- What are the point of discussion in Discussion section? Is is focused on the limitations of the present technologies and solution suggestions? Each of the six subsections have been written in different directions, but it is hard to determine and distinguish the intent of the writing, especially compared to Result section. If the authors have tried to illustrate the limitations of the state-of-the-art AI algorithms of previous studies, which are entirely dealt with in Results section, the structure of the manuscript should be modified. I recommend the authors reconstruct Result and Discussion sections for efficient communication of the authors' intention. There is no need for the structure of a review article to be identical to that of an original research article.
Minor points
- Please check and correct typos. For example, a vertical bar next to 'Abstract' of Line 11 seems to be a typo. Feb 10, 2024.Review of Line 90 also should be checked. I'll not designate the others further.
- Please standardize the notation of references. For example, from Line 52 to 61 and from Line 74 to 79 show inconsistent notation of the references, and I couldn't find them from Reference section. If the references are essential in this study, please add them to the Reference section. I'll not designate the other citations.
Response to Reviewer 3:
Thank you for your insightful and constructive comments. Kindly find, changes made to reflect on them respectively:
Major Points (changes made to address reviewer 3 comments are highlighted in yellow in the manuscript)
- There seems to have been an issue with the compilation of references and the reference manager (PaperPile) used for the manuscript draft like due to the conversion of the manuscript draft into the journal format template. These issues have been addressed with the appropriate references attributed in the introduction section (regarding the progression of the development of the technology as well as the standardization of the citation format throughout the document).
- We added a transition sentence into last paragraph of the introduction section to introduce ROSE process as we referred to ROSE studies in the lung section and we wanted to highlight the AI-empowered ROSE process as an advancement in the field for the readers.
In page 2, lines 72-73, added “Additionally, we will emphasize that rapid onsite evaluation (ROSE) of cytology specimens, enhanced by AI, presents a transformative opportunity in the field.”
In page 2, lines 87-91, added “AI-empowered ROSE holds the promise of delivering immediate feedback, ensuring better adequacy assessments, and streamlining workflows. Incorporating AI into ROSE processes has the potential to revolutionize point-of-care diagnostics by significantly reducing turnaround times and improving accuracy.”
- We have added direct references to the figures in the body of the methods section
- We appreciate the reviewer's thoughtful and constructive comment. The paragraph in question in section 3.1 was designed to complement Table 1 by elaborating on its contents rather than duplicating them. While the table provides a concise summary of the studies' key characteristics—such as algorithms, datasets, and performance metrics—the paragraph contextualizes these findings in a chronological framework. We purposefully chose this chronological structure to highlight the progression of methodologies, advancements in neural network applications, and how these developments have addressed (or failed to address) specific challenges over time.
Additionally, some studies listed in Table 1 are from the same research groups but conducted in different years. By discussing these works sequentially, we sought to illustrate how the same authors adapted their methodologies and achieved different outcomes over time. For example, Gopinath et al.'s progression from a single classifier approach in 2013 to a combined classifier ensemble in 2015 reflects their ongoing efforts to enhance diagnostic accuracy and reliability.
The paragraph also aims to synthesize insights beyond performance metrics, including methodological limitations and specific challenges unique to thyroid cytopathology. For instance, studies by Hirokawa et al. and Zhu et al. illustrate limitations related to dataset size, cross-validation, and benchmarking, which are pivotal for understanding the generalizability and real-world applicability of AI techniques in this domain.
We thank and acknowledge the reviewer's suggestion to categorize the studies by study design and neural network types and understand the rationale behind it. However, we believe that restructuring the paragraph in this way would detract from our intent to highlight the chronological progression of methodologies and the rationale behind the evolution of these studies. This chronological narrative underscores how advancements in the field have built upon prior work and adapted to address emerging challenges, which is central to the message of this section. While we will revisit the paragraph to ensure clarity and coherence, we aim to preserve the focus on the historical development and interconnectedness of the studies to effectively convey their collective significance.
We thank and appreciate the reviewer's thoughtful feedback on the coherence and direction of the discussion section. Based on this feedback, we have reordered and structured the discussion section to enhance clarity and align with the intent of the manuscript. This revised structure groups related themes for improved coherence and narrative flow, as suggested by the reviewer. Additionally, we aimed to emphasize both the successes and their impacts on advancing cytopathology while identifying pitfalls that future research can address. The discussion section provides actionable insights into building upon current advancements and circumventing challenges. Furthermore, we have incorporated considerations from the FDA's SaMD white paper to highlight the importance of aligning AI applications in cytopathology with evolving regulatory frameworks to ensure compliance.
Minor Points:
We thank the reviewer for their informative and constructive feedback. The document has been revised, and spelling/format has been corrected.
Round 2
Reviewer 2 Report
Comments and Suggestions for Authors
We thank the authors for addressing our previous comments.
Author Response
Thanks to the reviewer.
Reviewer 3 Report
Comments and Suggestions for Authors
To the authors
The authors have revised the manuscript based on the revision comments. However, the revised manuscript and response don't seem to be well-edited. I highly recommend the authors reflect the feedbacks and improve the whole manuscript.
1. This manuscript could have been upgraded only through relevant rewriting. The subsections and paragraphs, generally speaking, might not be modified without the restructuring manuscript, which I recommended. It doesn't mean merely adding some text because the intended logic only could be revealed through the structure of the writing.
2. Please make the reponse letter detailed. If the authors revised specific parts, it should be clearly indicated line-by-line in the response letter. It will help reviewers recognize which parts have been modified based on which feedbacks. Unless, it might be very hard for reviewers to figure out.
3. Revising the manuscript elaborately, at least in part, might be a better option rather than preserving the original text for revision process.
Author Response
We sincerely thank the reviewer for their thoughtful feedback. We have carefully considered and addressed each of their comments, as detailed below, and made corresponding changes to the manuscript. While we appreciate the reviewer's observations, we respectfully disagree with the second round of comments, as we believe the revisions clearly reflect the suggested improvements. In its current form, we consider the manuscript to be original and comprehensive, addressing the key challenges and opportunities in cytology AI. We also acknowledge and respect the possibility of differing perspectives on the matter.
Comments from reviewer 3
The authors conducted a systematic review of artificial intelligence related to cytopathology. The authors searched hundreds of scientific papers from PubMed DB within the last 5 years, and segemented previous studies into three categories based on the organs. They reviewed the cases, algorithms, performances, and limitations of each reference. As a review article, selection process of the studies seems to be relevant. However, the authors should address the issues commented as below.
Major points
- Review articles should cover diverse accumulated reference, considering the passage of time and adavancement in technology. When the authors develop the logics of the study, almost every point of information would need appropriate references. However, the authors cited only one reference by Trisolini, R. et al. from Reference section in Introduction section, and the other sections, especially Discussion section, also have scarce references, relatively. It would be the most critical issue for the authors to address, and it might require a modification of the entire manuscript.
- The structure of Introduction section and the intention of 'Rapid Onsite Evaluation (ROSE) of Cytology Specimens' subsection are confusing. The caption text of Fig 1 should also show sufficient and detailed contents. Although the authors wrapped up Introduction and suggest the purpose of the study, what would be this subsection for? I recommend the authors reconstruct the Introduction section, or move the subsection to Methods section.
- Figures in this study lack of relevant explation, and even Fig 2, 3 are not commented in the main text. If the figures are essential for this study, the authors should make relevant linkages between the main text and the figures, and manifest them to the readers.
- The last paragraph of 3.1 Thyroid section occupies a full page, and the it has simply enumerated the number of data, adopted algorithms, performance, and limitation of each reference. This might be just another information of Table 1, and lack readability. I recommend the authors take a closer look into the references, and categorize them according to the study design and the types of neural networks. It seems that the same work is need in the other sections as well.
- What are the point of discussion in Discussion section? Is is focused on the limitations of the present technologies and solution suggestions? Each of the six subsections have been written in different directions, but it is hard to determine and distinguish the intent of the writing, especially compared to Result section. If the authors have tried to illustrate the limitations of the state-of-the-art AI algorithms of previous studies, which are entirely dealt with in Results section, the structure of the manuscript should be modified. I recommend the authors reconstruct Result and Discussion sections for efficient communication of the authors' intention. There is no need for the structure of a review article to be identical to that of an original research article.
Minor points
- Please check and correct typos. For example, a vertical bar next to 'Abstract' of Line 11 seems to be a typo. Feb 10, 2024.Review of Line 90 also should be checked. I'll not designate the others further.
- Please standardize the notation of references. For example, from Line 52 to 61 and from Line 74 to 79 show inconsistent notation of the references, and I couldn't find them from Reference section. If the references are essential in this study, please add them to the Reference section. I'll not designate the other citations.
Response to Reviewer 3:
Thank you for your insightful and constructive comments. Kindly find, changes made to reflect on them respectively:
Major Points (changes made to address reviewer 3 comments are highlighted in yellow in the manuscript)
- There seems to have been an issue with the compilation of references and the reference manager (PaperPile) used for the manuscript draft like due to the conversion of the manuscript draft into the journal format template. These issues have been addressed with the appropriate references attributed in the introduction section (regarding the progression of the development of the technology as well as the standardization of the citation format throughout the document).
- We added a transition sentence into last paragraph of the introduction section to introduce ROSE process as we referred to ROSE studies in the lung section and we wanted to highlight the AI-empowered ROSE process as an advancement in the field for the readers.
In page 2, lines 72-73, added “Additionally, we will emphasize that rapid onsite evaluation (ROSE) of cytology specimens, enhanced by AI, presents a transformative opportunity in the field.”
In page 2, lines 87-91, added “AI-empowered ROSE holds the promise of delivering immediate feedback, ensuring better adequacy assessments, and streamlining workflows. Incorporating AI into ROSE processes has the potential to revolutionize point-of-care diagnostics by significantly reducing turnaround times and improving accuracy.”
- We have added direct references to the figures in the body of the methods section
- We appreciate the reviewer's thoughtful and constructive comment. The paragraph in question in section 3.1 was designed to complement Table 1 by elaborating on its contents rather than duplicating them. While the table provides a concise summary of the studies' key characteristics—such as algorithms, datasets, and performance metrics—the paragraph contextualizes these findings in a chronological framework. We purposefully chose this chronological structure to highlight the progression of methodologies, advancements in neural network applications, and how these developments have addressed (or failed to address) specific challenges over time.
Additionally, some studies listed in Table 1 are from the same research groups but conducted in different years. By discussing these works sequentially, we sought to illustrate how the same authors adapted their methodologies and achieved different outcomes over time. For example, Gopinath et al.'s progression from a single classifier approach in 2013 to a combined classifier ensemble in 2015 reflects their ongoing efforts to enhance diagnostic accuracy and reliability.
The paragraph also aims to synthesize insights beyond performance metrics, including methodological limitations and specific challenges unique to thyroid cytopathology. For instance, studies by Hirokawa et al. and Zhu et al. illustrate limitations related to dataset size, cross-validation, and benchmarking, which are pivotal for understanding the generalizability and real-world applicability of AI techniques in this domain.
We thank and acknowledge the reviewer's suggestion to categorize the studies by study design and neural network types and understand the rationale behind it. However, we believe that restructuring the paragraph in this way would detract from our intent to highlight the chronological progression of methodologies and the rationale behind the evolution of these studies. This chronological narrative underscores how advancements in the field have built upon prior work and adapted to address emerging challenges, which is central to the message of this section. While we will revisit the paragraph to ensure clarity and coherence, we aim to preserve the focus on the historical development and interconnectedness of the studies to effectively convey their collective significance.
We thank and appreciate the reviewer's thoughtful feedback on the coherence and direction of the discussion section. Based on this feedback, we have reordered and structured the discussion section to enhance clarity and align with the intent of the manuscript. This revised structure groups related themes for improved coherence and narrative flow, as suggested by the reviewer. Additionally, we aimed to emphasize both the successes and their impacts on advancing cytopathology while identifying pitfalls that future research can address. The discussion section provides actionable insights into building upon current advancements and circumventing challenges. Furthermore, we have incorporated considerations from the FDA's SaMD white paper to highlight the importance of aligning AI applications in cytopathology with evolving regulatory frameworks to ensure compliance.
Minor Points:
We thank the reviewer for their informative and constructive feedback. The document has been revised, and spelling/format has been corrected.
Round 3
Reviewer 3 Report
Comments and Suggestions for Authors
The authors have revised the manuscript based on the revision comments. However, the revised manuscript and response don't seem to be well-edited. I highly recommend the authors reflect the feedbacks and improve the whole manuscript.
1. This manuscript could have been upgraded only through relevant rewriting. The subsections and paragraphs, generally speaking, might not be modified without the restructuring manuscript, which I recommended. It doesn't mean merely adding some text because the intended logic only could be revealed through the structure of the writing.
2. Please make the reponse letter detailed. If the authors revised specific parts, it should be clearly indicated line-by-line in the response letter. It will help reviewers recognize which parts have been modified based on which feedbacks. Unless, it might be very hard for reviewers to figure out.
3. Revising the manuscript elaborately, at least in part, might be a better option rather than preserving the original text for revision process.
Author Response
The authors conducted a systematic review of artificial intelligence related to cytopathology. The authors searched hundreds of scientific papers from PubMed DB within the last 5 years, and segemented previous studies into three categories based on the organs. They reviewed the cases, algorithms, performances, and limitations of each reference. As a review article, selection process of the studies seems to be relevant. However, the authors should address the issues commented as below.
Major points:
- Review articles should cover diverse accumulated reference, considering the passage of time and adavancement in technology. When the authors develop the logics of the study, almost every point of information would need appropriate references. However, the authors cited only one reference by Trisolini, R. et al. from Reference section in Introduction section, and the other sections, especially Discussion section, also have scarce references, relatively. It would be the most critical issue for the authors to address, and it might require a modification of the entire manuscript.
- The structure of Introduction section and the intention of 'Rapid Onsite Evaluation (ROSE) of Cytology Specimens' subsection are confusing. The caption text of Fig 1 should also show sufficient and detailed contents. Although the authors wrapped up Introduction and suggest the purpose of the study, what would be this subsection for? I recommend the authors reconstruct the Introduction section, or move the subsection to Methods section.
- Figures in this study lack of relevant explation, and even Fig 2, 3 are not commented in the main text. If the figures are essential for this study, the authors should make relevant linkages between the main text and the figures, and manifest them to the readers.
- The last paragraph of 3.1 Thyroid section occupies a full page, and the it has simply enumerated the number of data, adopted algorithms, performance, and limitation of each reference. This might be just another information of Table 1, and lack readability. I recommend the authors take a closer look into the references, and categorize them according to the study design and the types of neural networks. It seems that the same work is need in the other sections as well.
- What are the point of discussion in Discussion section? Is is focused on the limitations of the present technologies and solution suggestions? Each of the six subsections have been written in different directions, but it is hard to determine and distinguish the intent of the writing, especially compared to Result section. If the authors have tried to illustrate the limitations of the state-of-the-art AI algorithms of previous studies, which are entirely dealt with in Results section, the structure of the manuscript should be modified. I recommend the authors reconstruct Result and Discussion sections for efficient communication of the authors' intention. There is no need for the structure of a review article to be identical to that of an original research article.
Minor points:
1. Please check and correct typos. For example, a vertical bar next to 'Abstract' of Line 11 seems to be a typo. Feb 10, 2024.Review of Line 90 also should be checked. I'll not designate the others further.
2. Please standardize the notation of references. For example, from Line 52 to 61 and from Line 74 to 79 show inconsistent notation of the references, and I couldn't find them from Reference section. If the references are essential in this study, please add them to the Reference section. I'll not designate the other citations.
--------------------------------------------------------------------------
Response: Thank you for your insightful and constructive comments. Kindly find, changes made to reflect on them respectively:
Major Points (changes made to address reviewer 3 comments are highlighted in yellow in the manuscript)
There seems to have been an issue with the compilation of references and the reference manager (PaperPile) used for the manuscript draft like due to the conversion of the manuscript draft into the journal format template. These issues have been addressed with the appropriate references attributed in the introduction section (regarding the progression of the development of the technology as well as the standardization of the citation format throughout the document). We added a transition sentence into last paragraph of the introduction section to introduce ROSE process as we referred to ROSE studies in the lung section and we wanted to highlight the AI-empowered ROSE process as an advancement in the field for the readers.
In page 2, lines 72-73, added "Additionally, we will emphasize that rapid onsite evaluation (ROSE) of cytology specimens, enhanced by AI, presents a transformative opportunity in the field."
In page 2, lines 87-91, added "AI-empowered ROSE holds the promise of delivering immediate feedback, ensuring better adequacy assessments, and streamlining workflows. Incorporating AI into ROSE processes has the potential to revolutionize point-of-care diagnostics by significantly reducing turnaround times and improving accuracy."
We have added direct references to the figures in the body of the methods section We appreciate the reviewer's thoughtful and constructive comment. The paragraph in question in section 3.1 was designed to complement Table 1 by elaborating on its contents rather than duplicating them. While the table provides a concise summary of the studies' key characteristics-such as algorithms, datasets, and performance metrics-the paragraph contextualizes these findings in a chronological framework. We purposefully chose this chronological structure to highlight the progression of methodologies, advancements in neural network applications, and how these developments have addressed (or failed to address) specific challenges over time.
Additionally, some studies listed in Table 1 are from the same research groups but conducted in different years. By discussing these works sequentially, we sought to illustrate how the same authors adapted their methodologies and achieved different outcomes over time. For example, Gopinath et al.'s progression from a single classifier approach in 2013 to a combined classifier ensemble in 2015 reflects their ongoing efforts to enhance diagnostic accuracy and reliability.
The paragraph also aims to synthesize insights beyond performance metrics, including methodological limitations and specific challenges unique to thyroid cytopathology. For instance, studies by Hirokawa et al. and Zhu et al. illustrate limitations related to dataset size, cross-validation, and benchmarking, which are pivotal for understanding the generalizability and real-world applicability of AI techniques in this domain.
We thank and acknowledge the reviewer's suggestion to categorize the studies by study design and neural network types and understand the rationale behind it. However, we believe that restructuring the paragraph in this way would detract from our intent to highlight the chronological progression of methodologies and the rationale behind the evolution of these studies. This chronological narrative underscores how advancements in the field have built upon prior work and adapted to address emerging challenges, which is central to the message of this section. While we will revisit the paragraph to ensure clarity and coherence, we aim to preserve the focus on the historical development and interconnectedness of the studies to effectively convey their collective significance.
We thank and appreciate the reviewer's thoughtful feedback on the coherence and direction of the discussion section. Based on this feedback, we have reordered and structured the discussion section to enhance clarity and align with the intent of the manuscript. This revised structure groups related themes for improved coherence and narrative flow, as suggested by the reviewer. Additionally, we aimed to emphasize both the successes and their impacts on advancing cytopathology while identifying pitfalls that future research can address. The discussion section provides actionable insights into building upon current advancements and circumventing challenges. Furthermore, we have incorporated considerations from the FDA's SaMD white paper to highlight the importance of aligning AI applications in cytopathology with evolving regulatory frameworks to ensure compliance.
Minor Points:
We thank the reviewer for their informative and constructive feedback. The document has been revised, and spelling/format has been corrected.